# Antimicrobial Activity of *Ligilactobacillus animalis* SWLA-1 and Its Cell-Free Supernatant against Multidrug-Resistant Bacteria and Its Potential Use as an Alternative to Antimicrobial Agents

**DOI:** 10.3390/microorganisms11010182

**Published:** 2023-01-11

**Authors:** Hong-Jae Lee, Joong-Bok Lee, Seung-Yong Park, In-Soo Choi, Sang-Won Lee

**Affiliations:** Laboratory of Infectious Diseases and Veterinary Microbiology, College of Veterinary Medicine, Konkuk University, Seoul 05029, Republic of Korea

**Keywords:** *Ligilactobacillus animalis*, antimicrobial substances, multidrug-resistant (MDR) bacteria, cell-free supernatant (CFS), competitive coculture assay, spot agar assay, spot inhibition index (SII)

## Abstract

The emergence of multidrug-resistant (MDR) bacteria and the spread of antimicrobial resistance among various bacteria are major threats to the global community. Due to the increased failure of classical antibiotic treatments against MDR bacterial infections, probiotics and their antimicrobial compounds have been suggested as promising alternatives to deal with MDR bacteria. Various strains of lactic acid bacteria have been reported to produce antagonistic molecules against pathogens. A new strain of *Ligilactobacillus animalis*, *L. animalis* SWLA-1, isolated from the feces of healthy dogs, shows strong antimicrobial activity against not only common pathogens but also MDR bacteria. In this study, we compared the antimicrobial activity of *L. animalis* SWLA-1 with that of other lactobacilli and antibiotics using an agar spot assay. Additionally, a novel spot inhibition index was developed and validated to quantitively evaluate the inhibitory activities of lactobacilli and antibiotics. A competitive coculture assay of *L. animalis* SWLA-1 with MDR bacteria further demonstrated its antibacterial activity. Furthermore, we evaluated the antimicrobial activity of the cell-free supernatant (CFS) of *L. animalis* SWLA-1 and its stability under various conditions in vitro. We found that *L. animalis* SWLA-1 and its CFS are potential alternatives to classic antimicrobial agents.

## 1. Introduction

Lactic acid bacteria (LAB) have a long history of symbiosis with humans and animals [1,2,3]. The *Lactobacillus* genus, a major genus of LAB, is well known for its beneficial functions in the host and for its production of various effective metabolites and derivatives. The applications of these bacteria range from starter cultures in milk supplements, vegetables, meat, and fermented foods in the food industry [4] to probiotics used in public healthcare [5] and the livestock industry [6].

In addition, other fields concerned with bacterial infection and public health investigate LAB because of their antimicrobial activity. Microorganisms, including *Lactobacillus*, have long competed fiercely with each other to secure nutrients and territory. To this end, various microorganisms have developed various survival tactics and defense mechanisms against others [7]. For example, prokaryotes secrete bacteriocins, which are ribosomally synthesized peptides that have antibacterial properties. In addition, LAB metabolites, including natural organic acids, are effective chemicals used to overwhelm other bacteria [8].

Since the first antibiotic, penicillin, was introduced via the pioneering work of André Gratia and Alexander Fleming in the 1920s [9], many antibiotics and disinfectants have been successively discovered and commercialized. Since then, antibiotic resistance has spread among common pathogens, such that global human and veterinary communities have faced the emergence of multidrug-resistant (MDR) bacteria. The abuse and misuse of antibiotics in medicine and agriculture aggravated the spread of antibiotic resistance, which occurred faster than new antibiotics could be introduced. A World Health Organization report warned that the global threat of antibiotic-resistant pathogens may lead to 10 million deaths annually by 2050 [10]. Due to treatment failures using classical antibiotics and the lack of newly developed antimicrobial agents, alternative strategies are being suggested to control bacterial infections. LAB and their antimicrobial derivatives are among the promising strategies [11,12].

In a previous study, several *Ligilactobacillus animalis* strains with antimicrobial activity against multiple pathogenic bacteria were isolated from the feces of healthy dogs [13]. *L. animalis* SWLA-1, which has significantly higher antibacterial activity than the other isolates, has been renamed *L. animalis* SWLA-1. The antimicrobial activity of *L. animalis* SWLA-1 against MDR bacteria was evaluated. *Lactobacillus plantarum* ATCC 14917 has strong antimicrobial activity against pathogenic bacteria and has been used as a reference strain in previous studies [14,15]. Although several studies have compared and evaluated the antimicrobial activity of lactobacilli using the size of the inhibition zone against indicator bacteria [14,16], the results do not consider various sizes of colony spots formed by lactobacilli.

In this study, we aimed to demonstrate the antimicrobial activity of *Ligilactobacillus animalis* SWLA-1, a newly discovered strain of *L. animalis* (formerly named *Lactobacillus animalis*), against pathogenic MDR bacteria isolated from human specimens and veterinary field samples. The antimicrobial activity of LAB against clinical pathogens was evaluated using competitive coculture and spot agar assays with reference strain groups; additionally, a novel spot inhibition index (SII) was developed. Considering that the size of the colony spot affects the diameter of the inhibition zone, the SII was calculated based on the ratio of the inhibition zone diameter to the colony spot diameter and used to quantitatively evaluate the antimicrobial activity of *L. animalis* SWLA-1. Furthermore, the SII was validated by comparing the inhibitory effects of three lactobacilli and two antibiotics (amikacin (AK) and ampicillin (AMP)). Additionally, the antimicrobial effect of the cell-free supernatant (CFS) of lactobacilli and its stability against various treatments were evaluated.

## 2. Materials and Methods

### 2.1. Preparation of Bacterial Strains

The antimicrobial activity of *L. plantarum* ATCC 14917 was compared with that of *L. animalis* SWLA-1, which was isolated from healthy dogs [13]. Another *L. animalis* strain isolated from healthy dogs, *L. animalis* 11-2, which had weak antimicrobial activity against indicator bacteria in a previous study [13], was also compared with that of other *Lactobacillus* strains. Frozen pure cultures of these lactobacilli were plated on Difco™ de Man Rogosa Sharpe (MRS) agar (BD Biosciences, Sparks, MD, USA) containing 0.5% lactose (*v*/*v*) [17]; each colony of *Lactobacillus* strains was cultured in MRS broth and used in every experiment.

MDR bacteria that were clinically isolated from animals and humans were used in this study. These bacteria were resistant to more than three antimicrobial agents. The antimicrobial susceptibility profiles of these indicator bacteria are provided in Appendix A (R = resistant, S = susceptible, and I = intermediate). The bacteria included *Salmonella enterica* serovar Gallinarum CNHJ001, *Salmonella enterica* serovar Enteritidis 190610_1, *Escherichia coli* ROH_0034, and *Staphylococcus aureus* ROH_0029. *Salmonella enterica* serovar Gallinarum CNHJ001 and *Salmonella enterica* serovar Enteritidis 190610_1 were isolated from chickens, and the other bacteria were isolated from human patients and provided by the Korea Disease Control and Prevention Agency (KDCPA). These indicator bacteria were primarily cultured on blood agar plates, and each bacterial colony was cultured in Bacto™ Tryptic Soy Broth (TSB) (BD Biosciences, Grenoble, France) and used in every experiment.

### 2.2. Effect of pH on Indicator Bacteria

Apart from antibacterial substances, the pH concentration of the environment surrounding bacteria affects bacterial growth [18,19]. To determine the optimal pH concentration range for the independent inhibitory activity of antimicrobial substances from lactobacilli against indicator bacteria without the effect of acid stress, each indicator bacteria was cultured under four different pH conditions. Modified TSB-MRS broth, containing twice as much concentrated 5 mL TSB and 5 mL MRS (*v*/*v*), was prepared to culture the indicator bacteria [20]. The TSB-MRS broth was acidified with 1 M hydrochloric acid to pH 6.5, 6.0, or 5.5. Each indicator strain was precultured in TSB (A_600_ 0.090–0.100) and diluted with sterile TSB to obtain 1.5 × 10^6^ colony-forming units (CFU)/mL. Thereafter, 100 µL of this diluted bacterial culture was inoculated in standard TSB-MRS (pH 6.8) and acidified TSB-MRS broths (pH 6.5, 6.0, 5.5, or 5.0). Bacterial growth assessment and colony counting on tryptic soy agar were conducted at time zero and at 4, 8, and 24 h. All CFU values were measured and logarithmically transformed (log_10_ CFU/mL) to compare the mean difference between the values at 0 h and each time point. The experiment was performed independently in triplicate.

### 2.3. Comparison of Antimicrobial Activity of L. animalis SWLA-1 with That of Other Lactobacilli

#### 2.3.1. Comparative Agar Spot Assay

The antimicrobial activities of *L. plantarum* ATCC 14917, *L. animalis* SWLA-1, and *L. animails* 11-2 were compared on the MRS plate. Since all indicator bacteria were susceptible to AK and resistant to AMP, AK and AMP antibiotic disks (Oxoid, Cheshire, UK) were used as references in this experiment. The three bacteria were incubated in MRS broth (37 °C for 4 h, 200 rpm), and 3 µL of each broth culture (A_600_ 0.500) was spot-inoculated on each section of the MRS agar plate. Each spot contained 1.5 × 10^5^ CFU. Next, an equal volume of phosphate-buffered saline (PBS) was inoculated on another section of the MRS agar plate. Then, the AK and AMP disks were placed on the other sections of the same plate, after which the plates with inoculated bacterial spots were incubated at 37 °C for 24 h. Subsequently, the bacterial culture of the indicator bacteria (≈2 × 10^9^ CFU/mL) was mixed with 10 mL of soft Muller–Hinton agar (0.8%) and overlaid on the Lactobacillus-spotted MRS agar plate. After spotting onto the solidified soft agar overlay, all plates were incubated at 37 °C for 24 h. Ten plate copies were tested per indicator bacteria, and the longest and shortest diameters of the inhibitory zone were measured. Although the same volume and CFU of each bacterial culture were inoculated on the plate, various sizes of lactobacilli spots were formed. Therefore, the spot inhibition index (SII) was calculated as the average of the longest diameter of the inhibitory zone divided by the longest diameter of the Lactobacillus spot and the shortest diameter of the inhibitory zone divided by the shortest diameter of the Lactobacillus spot, as shown in Figure 1. This study adopted this novel method to quantitively compare the antimicrobial activity of *L. animalis* SWLA-1 to those of the other lactobacilli. Due to the even diameter of the antibiotic disks, the longest and shortest diameters of the inhibition zones created by AK and AMP were divided by the equal diameter of the disk.

#### 2.3.2. Comparative Coculture Assay

The antimicrobial activity of three lactobacilli strains was compared using a coculture assay with modified TSB-MRS broth. Each lactobacillus strain was precultured in MRS broth (A_600_ 0.090–0.100) and diluted with sterile MRS to obtain 1.5 × 10^6^ CFU/mL. Each indicator bacterium was also preculred in TSB (A_600_ 0.090–0.100) and diluted with sterile TSB to obtain a CFU count equivalent to that of Lactobacillus CFU (~1.5 × 10^6^ CFU/mL). Subsequently, 100 µL of the diluted Lactobacillus and pathogen culture were inoculated into 9.8 mL of TSB-MRS broth and cocultured. Afterward, the bacterial culture was collected and plated on selective media to count the CFU of the indicator bacteria and lactobacilli at 0, 4, 8, and 24 h. The pH concentration of the culture was also measured at each time point. Salmonella Chromo Select Agar (Sigma-Aldrich, St. Louis, MO, USA) was used to selectively count colonies of *S. Gallinarum*, *S. Enteritidis*, and *E. coli*. Staphylococcus medium No. 110 (Oxoid) was used for selective isolation and colony counting of *S. aureus*. The pH of the MRS agar was adjusted to 5.0 in order to inhibit the growth of other cocultured pathogenic bacteria and selectively count the Lactobacillus colonies. Growth in TSB-MRS broth inoculated only with indicator bacteria was compared with that in the experimental groups as a reference for normal bacterial growth. All CFU values were measured and logarithmically transformed (log_10_ CFU/mL) to compare the mean differences between the groups. The experiment was performed independently in triplicate.

### 2.4. Antimicrobial Activity of Supernatant Derived from L. animalis SWLA-1

#### 2.4.1. Preparation of Cell-Free Supernatant

An overnight culture of *L. animalis* SWLA-1 in MRS broth (≈1.2 × 10^9^ CFU/mL) was collected and centrifuged at 10,000× *g* and 4 °C for 30 min (Legend X1R; Thermo Fisher Scientific, Waltham, MA, USA). The centrifuged supernatant was collected in 50 mL conical tubes and filtered with a 0.2 µm pore filter system. This collected supernatant was used to ascertain whether any viable cells remained on the MRS plate. Subsequently, some of this cell-free supernatant (CFS) was 10-fold concentrated by using lyophilization (FreeZone Plus 12 L Cascade Console Freeze Dry System; Labconco, Kansas City, MO, USA) and resuspended in PBS.

#### 2.4.2. Various Treatments and Effects on Antimicrobial Activity of CFS

The CFS of *L. animalis* SWLA-1 was treated with chemicals or enzymes to evaluate changes in its antimicrobial activity under various conditions. To evaluate thermostability, the CFS was heated at 40, 60, and 80 °C for 2 h and at 100 °C for 30 min. Next, each CFS sample was cooled to room temperature and aliquoted into a 96-microwell plate using two-fold microdilution. To evaluate the acid tolerance of the CFS, each CFS sample was exposed to pH 2.0, 4.0, 6.0, and 8.0; the pH was adjusted using 1 M hydrochloric acid or sodium hydroxide and incubated in a thermoshaker (900 rpm, 37 °C) for 2 h. Afterward, each CFS sample was readjusted to pH 6.8 and two-fold diluted in a 96-well microwell plate. Protease treatments were also performed. Thereafter, proteinase K (2 mg/mL) or trypsin (1 mg/mL) was added to the CFS sample, and a proteolytic reaction was performed for 2 h in a thermoshaker (900 rpm, 37 °C). Each enzyme-treated CFS sample was then inactivated in a dry bath (60 °C, 5 min). Subsequently, the CFS samples were two-fold diluted in a 96-well plate. The original CFS of *L. animalis* SWLA-1 (derived from ≈1.2 × 10^9^ CFU/mL) and 10-fold-concentrated CFS of *L. animalis* SWLA-1 were also two-fold diluted in a 96-well plate for comparison with the treated CFS samples. Antimicrobial activity was evaluated using the prepared microdilution well plates for the four indicator bacteria, namely, *S. Gallinarum* CNHJ001, *S. Enteritidis* 190610_1, *E. coli* ROH_0034, and *S. aureus* ROH_0029. Each bacterium was diluted in cation-adjusted MH broth containing TES, and 50 µL of each was inoculated into a 96-well plate. The volume of the bacterial inoculum was the same as that of the aliquoted CFS samples. MRS and MH broths were equally mixed and used as the negative control. The experiment was performed independently in triplicate.

### 2.5. Statistical Analysis

The quantitative data of the experimental results are presented as mean ± standard deviation. Because the data were not normally distributed, the Kruskal–Wallis test was used to nonparametrically compare the multiple means of the experimental data. Thereafter, a post hoc Dunn’s test was used to compare the experimental data of each group. In the case of the competitive coculture assay, a two-way ANOVA test was also used to analyze the relations between the mean viable counts of indicator bacteria and two variables (type of LAB and pH). A post hoc Tukey’s test was used to compare the experimental data of each group. These tests were performed using the “rstatix” package in R (version 0.6.0; R Foundation for Statistical Computing, Vienna, Austria; Alboukadel Kassambara, 2020).

The effect and interaction of factors related to bacterial growth were determined using response-surface analysis. The pH and incubation time were the independent variables. Viable cell counts of indicator bacteria (log_10_ CFU/mL) were the response variables. This method was performed using the “rsm” package in R (version 2.10.3, Response-Surface Methods in R, Iowa City, IA, USA, Russell V. Lenth, 2009).

Significance was determined at an α level of 0.05 for all experiments.

## 3. Results

### 3.1. Effect of pH on Indicator Bacteria

The result for bacterial growth under various pH conditions is shown in Appendix A. Using the response-surface methodology, the effects of pH and incubation time on the viable counts of indicator bacteria were analyzed and are visualized in Figure 2. The effects of pH and time on bacterial growth were statistically significant (*p* < 0.05). The stationary points of pH were lower than 5.2 in all experimental groups. All groups of indicator bacteria incubated at pH 6.5 could grow as much as the group incubated at a standard pH concentration (6.8). In detail, *S. gallinarum* CNHJ001 showed no significant differences in growth between each pH condition until the 4 h point. After 8 h of incubation, the viable bacterial counts at pH 6.0 and 5.5 were significantly lower than those at pH 6.8 and 6.5 (*p* < 0.05). Significant bacterial growth inhibition was also observed after 24 h at pH 6.0 and 5.5 compared with that at pH 6.8 and 6.5 (*p* < 0.05). The inhibitory effect against *S. enteritidis* 190610_1 growth was observed at 4 and 8 h at pH 6.0 and 5.5 (*p* < 0.05). Inhibition was also observed after 24 h of incubation at pH 5.5 (*p* < 0.01), but *S. enteritidis* 190610_1 growth was not affected at pH 6.0 compared with that at pH 6.8 and 6.5. In the case of *E. coli* ROH_0034, growth was inhibited at pH 5.5 at each time point (4, 8, and 24 h) compared with that at pH 6.8 and 6.5 (*p* < 0.05), while it was significantly inhibited only at 8 h at pH 6.0. The growth of *S. aureus* ROH_0029 was significantly inhibited at pH 6.0 and 5.5 at 8 h compared with that under the other pH conditions (*p* < 0.05). *S. aureus* ROH_0029 growth was inhibited at 24 h only at a pH of 5.5 (*p* < 0.01).

### 3.2. Comparative Agar Spot Assay

The inhibitory activities of lactobacilli and the antibiotic disks against the four indicator bacteria are shown in Figure 3. The SII of each Lactobacillus or antibiotic disk was calculated and compared; the similarities and differences are shown in Figure 4. The SII of *L. animalis* SWLA-1 against all indicator bacteria was significantly higher than that of *L. animalis* 11-2 or AMP and was as high as that of AK, to which all indicator bacteria were susceptible (*p* < 0.05). The SII of *L. animalis* SWLA-1 against *S. gallinarum* CNHJ001 significantly differed only from that of *L. animalis* 11-2 compared with that of AK or *L. plantarum* ATCC 14917. The SII of *L. plantarum* ATCC 14917 against E. coli ROH_0034 or *S. aureus* ROH_0029 was significantly higher than that of *L. animalis* 11-2 or AMP (*p* < 0.05), but no significant difference was observed against *S. gallinarum* CNHJ001 and *S. Enteritidis* 190610_1 compared with that against *L. animalis* 11-2.

### 3.3. Comparative Coculture Assay

The results of culturing indicator bacteria independently or with each Lactobacillus are shown in Figure 5. All indicator bacteria cultured with lactobacilli showed reduced viable counts at each time point compared with those of the indicator bacteria cultured purely in the modified MRS-TSB medium, except for *S. enteritidis* 190610_1 and *L. animalis* 11-2 at 8 h. A significant reduction in the viable count was observed in every indicator bacterial group cultured with *L. animalis* SWLA-1 or *L. plantarum* ATCC 14917 compared with that in groups cultured without or with *L. animalis* 11-2 (*p* < 0.05). In the case of *S. enteritidis* 190610_1, all three lactobacilli were able to significantly inhibit the growth of this Salmonella strain compared with that of the solely cultured group at 4 h (*p* < 0.01).

The changes in pH and the corresponding viable count of each Lactobacillus in cocultured media are shown in Table 1. According to the results of the two-way ANOVA test, both the pH change and type of lactobacilli affected the growth of indicator bacteria (*p* < 0.01). Regardless of pH*, L. animalis* SWLA-1 and *L. plantarum* ATCC 14917 showed significant inhibitory activity against all indicator bacteria—*S. gallinarum* CNHJ001 (8 h), *S. enteritidis* 190610_1 (4 h, 8 h), *E. coli* ROH_0034 (8 h), and *S. aureus* ROH_0029 (8 h)—compared with the groups of indicator bacteria only and those cultured with *L. animalis* 11-2 (*p* < 0.01). At the 24 h time point, the growth of indicator bacteria in all experimental groups cultured with *L. animalis* SWLA-1 and *L. plantarum* ATCC 14917 was significantly inhibited compared with other groups and was affected by pH and the type of lactobacilli (*p* < 0.05). All groups of indicator bacteria cultured with lactobacilli showed reduced pH values of less than 5.0 after 24 h. There was no significant difference between the CFU of each Lactobacillus strain at each time point, except at 8 h. In all groups, *L. animalis* 11-2 showed lower viable counts compared with those of other lactobacilli at 8 h but reached CFU values similar to those of other lactobacilli at 24 h. *L. animalis* 11-2 cultured with *E. coli* ROH_0034 also showed lower viable counts than those of other lactobacilli at 4 h.

### 3.4. Effects of Various Treatments on CFS Antimicrobial Activity

The characteristic response of the CFS derived from *L. animalis* SWLA-1 to various treatments is shown in Table 2. The lowest concentration of the nontreated CFS, which inhibited bacterial growth, was determined at 100% antimicrobial activity. The lyophilized and 10-fold-concentrated CFS exhibited a 4-fold stronger inhibitory effect against all indicator bacteria relative to that of the crude CFS (1×). During thermostability experiments, the CFS sample was stable at temperatures between 40 and 60 °C in every experimental group. Unlike the results for E. coli ROH_0034 and *S. aureus* ROH_0029, reduced antimicrobial activities were observed in *S. Gallinarum* CNHJ001 and *S. Enteritidis* 190610_1 at 80 or 100 °C. The antimicrobial activity of the CFS was also affected by pH. In experimental groups, decreased inhibition efficiencies varied from as low as one-half to one-quarter. The enzymatic activity also reduced the CFS antimicrobial activity. Decreased antimicrobial activities due to trypsin were observed in all experimental groups, whereas the only decrease in activity due to proteinase K was observed in the *S. Enteritidis* group. The positive control, inoculated with each indicator in MRS + MH broth (pH 6.8), showed bacterial growth, whereas no growth was observed in the negative control.

## 4. Discussion

*Ligilactobacillus animalis* was first discovered by Dent and Williams [21] and named “*Lactobacillus animalis*,” which means “*Lactobacillus* in animals” in Latin. This bacterium has been isolated from various animals, such as baboons, patas monkeys, mice, and dogs. *Ligilactobacillus animalis* is a poorly understood bacterium compared with well-known lactobacilli such as *L. acidophilus*, *L. rhamnosus*, *L. reuteri*, or *L. plantarum*. However, several studies have used *L. animalis* as probiotics to improve the health of animals and prevent infection or as a novel food preservative [22,23,24]. Furthermore, a previous study discussed bacteriocin-producing *L. animalis* and conducted a characteristic analysis of the effects of its bacteriocin on pathogens in the fishing industry [25]. As a potential mechanism for countering the emerging threat of antimicrobial resistance, the antimicrobial activity of *L. animalis* SWLA-1 against MDR bacteria was investigated for the first time in this study.

In a competitive coculture assay, artificial conditions were created where the Lactobacilli could contact the pathogenic bacteria directly. We tried to figure out whether the Lactobacilli could inhibit the pathogenic bacteria and dominate the environment in these conditions. *L. animalis* SWLA-1 showed the highest *S*. *gallinarum* and *S*. *enteritidis* inhibition among the three lactobacilli. The anti-Salmonella activities of LAB strains have been reported by several authors [8,20,26], but it was demonstrated herein that *Lactobacillus* could inhibit other bacteria and dominate the medium when similar CFU quantities of *Lactobacillus* and *Salmonella* were inoculated. In addition, *L. animalis* SWLA-1 was able to inhibit *E. coli* and *S. aureus* at similar levels to those of *L. plantarum* ATCC 14917. Based on the results of inhibitory activity and the potential of *L. plantarum* ATCC 14917 as a biopreservative [14], *L. animalis* SWLA-1 could have possible applications as a preservative in food or start cultures. With regard to pH conditions and bacterial growth, the pH range of 6.8–6.5 did not affect the growth of the indicator bacteria in the present study. In particular, *L. animalis* SWLA-1 showed a significant inhibitory effect against *S*. *gallinarum* regardless of pH conditions in each experimental group (*p* < 0.01), while the growth of other indicator bacteria was affected by both variables (type of LAB and pH). Considering that the pH values of cocultured media between 0 h and 8 h ranged between 6.8 and 6.5, the significant inhibitory activity of *L. animalis* SWLA-1 against *S*. *gallinarum* resulted from other substances and not from acid stress. After 24 h of coculture, the pH dropped to less than 5.0. This made it difficult to evaluate the inhibitory activity of lactobacilli against the indicator bacteria without acid stress because, based on the results above, a pH of less than 5.5 is extremely harsh for bacterial strains.

The results of the agar spot assay showed that the inhibitory activity of *L. animalis* SWLA-1 against the clinical pathogens was comparable to that of *L. plantarum* ATCC 14917. All three lactobacilli used in this study can be classified as either strong or very strong inhibitors according to the classification made by Gaudana et al. [27]. Comparing the inhibitory activity of lactobacillus was challenging due to the various colonies and the similar sizes of the inhibition halos in the spot agar assay. By introducing the novel SII value in this study, it was possible to quantitively evaluate the inhibitory activity of each lactobacillus and compare them. The calculated SII focuses on the ratio of the inhibitory zone to the bacterial colony size, not the difference between the diameters of two quantities. Using the SII could provide a more precise comparison than using absolute differences between the diameters of the inhibitory zone and bacterial colony. In addition, the SII was validated by comparing the SIIs of lactobacilli to those of susceptible or resistant antibiotic disks. This suggests that the SII could be a quantitative method to compare the power of the inhibitory activity of lactobacilli against indicator bacteria. Combined with the results of the coculture assay, the inhibitory activity of *L. animalis* SWLA-1 was effective against *Salmonella* strains, which, due to its pathogen specificity, could be mainly observed in the antimicrobial activity of the bacteriocin, as described by Cotter et al. [28].

The antimicrobial characteristics of the CFS derived from *L. animalis* SWLA-1 were investigated in this study. Although several studies have reported antimicrobial substances in the CFS of lactobacilli [25,29,30], the present study further investigated the inhibitory effect of the treated CFS against each bacterium. In the cases of *E. coli* and *S. aureus*, only pH neutralization or trypsin treatment could affect the inhibitory effect of the CFS, showing constant antimicrobial activity in other conditions. In contrast, high temperatures, pH neutralization, and enzyme treatment caused a progressive decrease in the antimicrobial effect in both *Salmonella* strains. Reduced antimicrobial activity was also observed in the *E. coli* group compared with that in the other CFS treatment groups. Many substances in CFS, including bacteriocin, organic acids, and metabolites, can inhibit bacterial growth and have antimicrobial effects [29,30]. This indicates that the CFS of *L. animalis* SWLA-1 may include more than one antimicrobial substance, one being a protein or peptide and another a thermostable compound, allowing stability in a broad range of pH concentrations. It was proven that the antimicrobial characteristics of the CFS derived from *L. animalis* SWLA-1 were active after exposure to high temperatures, various pH concentrations, and proteolytic treatment in this study. Based on the results, this CFS could be sterilized for use as a disinfectant or endure the gastrointestinal environment when it is used as a feed additive.

To assess whether this antimicrobial activity is concentration-dependent, CFS lyophilization, which is an innovative method used to produce highly concentrated bacterial derivatives, was performed [31]. Although there was no precise correlation between the concentration and antimicrobial activity, the inhibitory effect of the CFS derived from *L. animalis* SWLA-1 affected the indicator bacterium in a concentration-dependent manner. Quantifying the antimicrobial substance in the CFS would be necessary to investigate its concentration-dependent characteristics.

Thus, further studies are required to identify this antimicrobial substance at a molecular level. It would be possible to precisely evaluate antimicrobial activity under various concentrations with the purification of this antimicrobial substance [25,32]. HPLC or LC/MS analysis is a favorable option for this investigation [25]. With the advances in sequencing technology, another approach would be to generate the complete genome of the target bacterium using high-throughput sequencing techniques (e.g., MinION or PacBio) and analyze its protein transcription or metabolic pathways.

## 5. Conclusions

In conclusion, this study suggests that *L. animalis* SWLA-1 and its CFS can significantly inhibit and regulate the growth of multidrug-resistant pathogenic bacteria, including both Gram-positive and Gram-negative species. With its antagonistic features against pathogenic bacteria, this versatile *Lactobacillus* showed its potential for use as a biocontrol strategy against pathogenic bacteria causing infectious diseases, such as fowl typhoid and human salmonellosis, and infections in animals and humans caused by *E. coli* or *Staphylococcus* spp. After decades of treatment methods against antibiotic resistance, this work contributes to the discovery of novel antimicrobial substances and alternatives to classical antibiotics. Furthermore, it lays a foundation for investigating beneficial bacteria that are not well-known but can act as reservoirs of antimicrobials against pathogens.

## Figures and Tables

**Figure 1 microorganisms-11-00182-f001:**
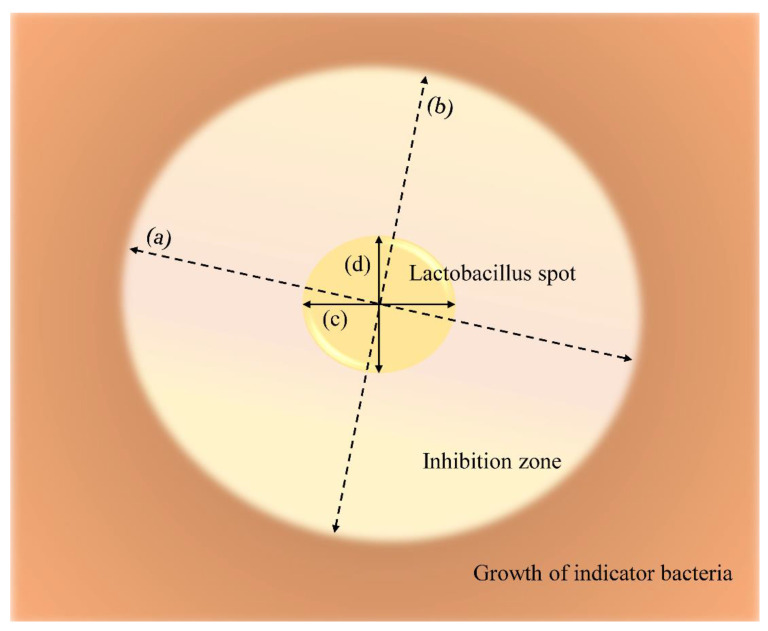
The definition of spot inhibition index (SII) and calculation method: (a) longest diameter of inhibitory zone, (b) shortest diameter of inhibitory zone, (c) longest diameter of lactobacillus spot, and (d) shortest diameter of lactobacillus spot. Each SII of lactobacillus was calculated as follows: Spot Inhibition Index (SII) = [M1] $\frac{1}{2}\times \left(\frac{longest\;diameter\;of\;inhibition\;zone}{longest\;diameter\;of\;lactobacillus\;spot}+\frac{shortest\;diameter\;of\;inhibition\;zone}{shortest\;diameter\;of\;lactobacillus\;spot}\right)$.

**Figure 2 microorganisms-11-00182-f002:**
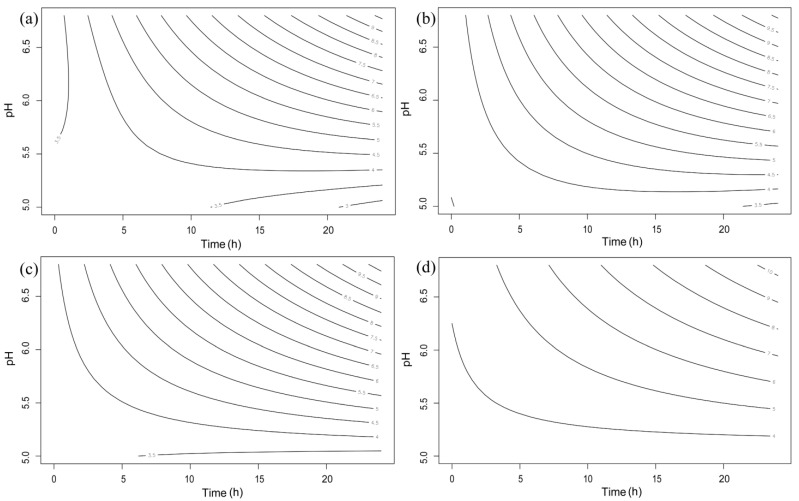
Fitted response-surface contour plots for the effect of pH and incubation time against the growth of four indicator bacteria: (**a**) SG: *Salmonella gallinarum* CNHJ001, (**b**) SE: *Salmonella enteritidis* 190610_1, (**c**) EC: *Escherichia coli* ROH_0034, and (**d**) SA: *Staphylococcus aureus* ROH_0029.

**Figure 3 microorganisms-11-00182-f003:**
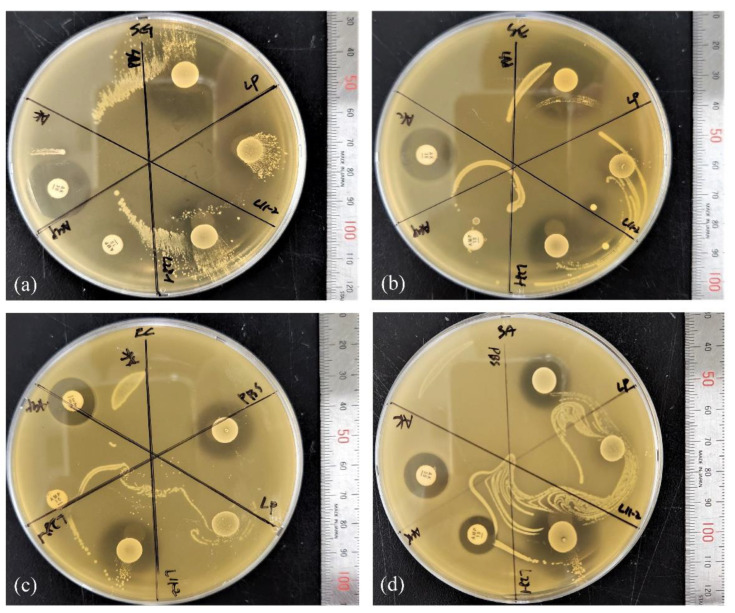
Agar spot assay with lactobacilli and antibiotic disks against the four indicator bacteria: (**a**) SG: *Salmonella gallinarum* CNHJ001, (**b**) SE: *Salmonella enteritidis* 190610_1, (**c**) EC: *Escherichia coli* ROH_0034, and (**d**) SA: *Staphylococcus aureus* ROH_0029. The sections are divided as follows: AK: amikacin; AMP: ampicillin; PBS: peptone-buffered water; LP: *L. plantarum* ATCC 14917; L28-1: *L. animalis* SWLA-1; L11-2: *L. animalis* 11-2.

**Figure 4 microorganisms-11-00182-f004:**
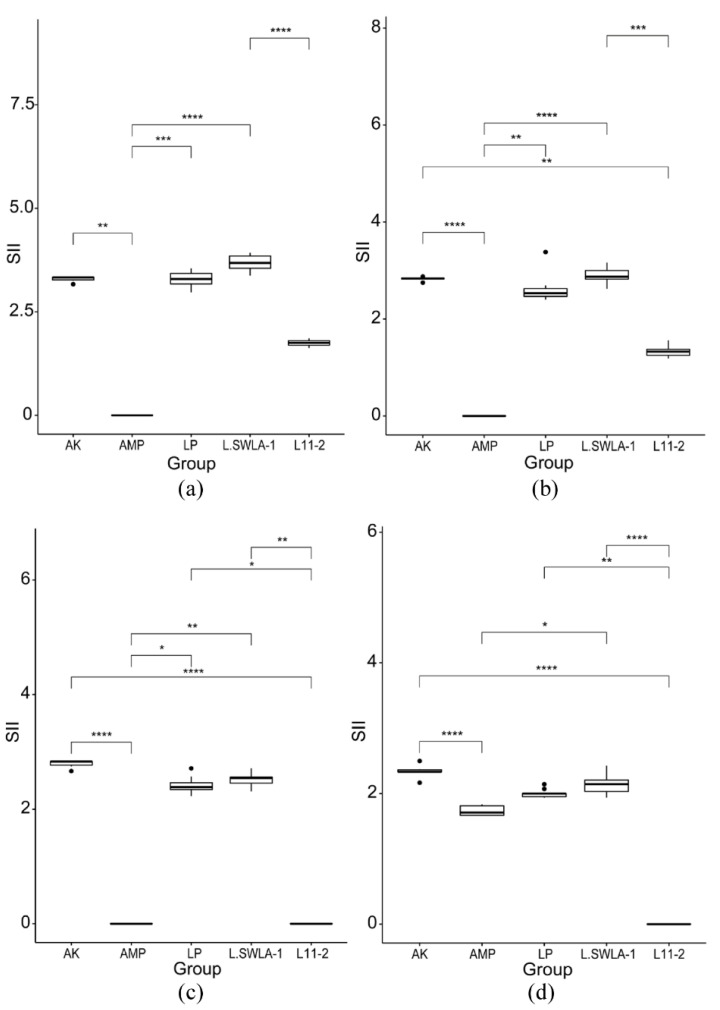
Differences in spot inhibition index (SII) values of the three lactobacilli and two antibiotics against the indicator bacteria: (**a**) *Salmonella gallinarum* CNHJ001, (**b**) *Salmonella enteritidis* 190610_1, (**c**) *Escherichia coli* ROH_0034, and (**d**) *Staphylococcus aureus* ROH_0029. The SII of each lactobacillus and antibiotic was calculated from ten replicates. The SII values of all groups were analyzed using a Kruskal–Wallis test followed by Dunn’s test for post hoc testing. Significant difference is denoted by asterisk * (*: *p* < 0.05, **: *p* < 0.01, ***: *p* < 0.001, ****: *p* < 0.0001). AK: amikacin; AMP: ampicillin; LP: *L. plantarum* ATCC 14917; L.SWLA-1: *L. animalis* SWLA-1; L11-2: *L. animalis* 11-2.

**Figure 5 microorganisms-11-00182-f005:**
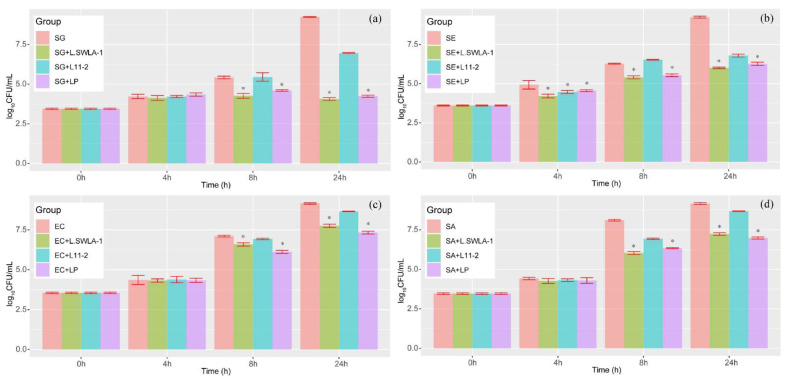
The results of the four indicator bacteria cocultured with each lactobacillus strain at each time point. Indicator bacterial strains are (**a**) SG: *Salmonella gallinarum* CNHJ001, (**b**) SE: *Salmonella enteritidis* 190610_1, (**c**) EC: Escherichia coli ROH_0034, and (**d**) SA: *Staphylococcus aureus* ROH_0029. The lactobacillus strains are L.SWLA-1: *L. animalis* SWLA-1, LP: *L. plantarum* ATCC 14917, and L11-2: *L. animalis* 11-2. Significant difference is denoted by asterisk * (*p* < 0.05).

**Table 1 microorganisms-11-00182-t001:** pH changes and viable counts of lactobacilli in the cocultured TSB-MRS broth at each time point.

Cocultured Bacteria	Time	pH Value and CFU of Lactobacilli Strains					
		*L. plantarum* ATCC 14917		*L. animalis* SWLA-1		*L. animalis* 11-2	
		pH	log_10_ CFU/mL	pH	log_10_ CFU/mL	pH	log_10_ CFU/mL
Salmonella gallinarum CNHJ001	0 h	6.80 ± 0.00	3.21 ± 0.04	6.80 ± 0.00	3.18 ± 0.06	6.80 ± 0.00	3.21 ± 0.04
	4 h	6.64 ± 0.01	4.39 ± 0.02	6.63 ± 0.05	4.10 ± 0.07	6.65 ± 0.02	4.25 ± 0.17
	8 h	6.55 ± 0.04	6.64 ± 0.05	6.58 ± 0.02	6.26 ± 0.15	6.60 ± 0.03	5.50 ± 0.22
	24 h	4.71 ± 0.04	9.21 ± 0.04	4.67 ± 0.01	9.16 ± 0.07	4.70 ± 0.03	9.09 ± 0.06
Salmonella enteritidis 190610_1	0 h	6.80 ± 0.00	3.22 ± 0.05	6.80 ± 0.00	3.22 ± 0.04	6.80 ± 0.00	3.11 ± 0.05
	4 h	6.61 ± 0.03	4.45 ± 0.05	6.65 ± 0.02	4.33 ± 0.03	6.64 ± 0.03	4.00 ± 0.08
	8 h	6.58 ± 0.04	6.96 ± 0.04	6.59 ± 0.01	6.59 ± 0.09	6.60 ± 0.02	5.08 ± 0.13
	24 h	4.55 ± 0.04	9.12 ± 0.04	4.53 ± 0.03	9.09 ± 0.04	4.52 ± 0.02	9.01 ± 0.50
Escherichia coli ROH_0034	0 h	6.80 ± 0.00	3.21 ± 0.07	6.80 ± 0.00	3.24 ± 0.06	6.80 ± 0.00	3.05 ± 0.03
	4 h	6.57 ± 0.01	4.40 ± 0.08	6.60 ± 0.02	4.44 ± 0.03	6.62 ± 0.04	3.41 ± 0.09
	8 h	6.53 ± 0.02	7.04 ± 0.04	6.55 ± 0.03	6.60 ± 0.09	6.60 ± 0.06	5.35 ± 0.13
	24 h	4.69 ± 0.03	9.09 ± 0.06	4.66 ± 0.01	8.96 ± 0.04	4.68 ± 0.02	9.09 ± 0.07
Staphylococcus aureus ROH_0029	0 h	6.80 ± 0.00	3.25 ± 0.03	6.80 ± 0.00	3.22 ± 0.06	6.80 ± 0.00	3.12 ± 0.05
	4 h	6.54 ± 0.02	4.32 ± 0.04	6.53 ± 0.03	4.46 ± 0.05	6.58 ± 0.02	4.00 ± 0.17
	8 h	6.33 ± 0.02	7.01 ± 0.06	6.31 ± 0.01	7.01 ± 0.06	6.28 ± 0.03	5.38 ± 0.13
	24 h	4.55 ± 0.01	9.15 ± 0.08	4.52 ± 0.02	9.11 ± 0.09	4.48 ± 0.01	9.01 ± 0.49

Time-dependent change in pH concentration in modified TSB-MRS broth at each time point. The pH value of the coculture medium was determined and recorded at time zero (0 h), 4 h, 8 h, and 24 h. The CFU of each lactobacillus in the coculture medium was also measured at the same time points.

**Table 2 microorganisms-11-00182-t002:** Different treatments on the CFS of *L. animalis* SWLA-1 and residual antimicrobial activity.

Treatment	Condition		Final pH	Residual Antimicrobial Activity (%)			
Concentration				*S*. *gallinarum*	*S. enteritidis*	*E. coli*	*S. aureus*
CFS (1×)	None		5.1	100 ± 0.0	100 ± 0.0	100 ± 0.0	100 ± 0.0
Concentrated (10×)	Lyophilized, 24 h		5.1	400 ± 0.0	400 ± 0.0	400 ± 0.0	400 ± 0.0
Temperature							
40 °C	Heated, 2 h		5.1	100 ± 0.0	100 ± 0.0	100 ± 0.0	100 ± 0.0
60 °C	Heated, 2 h		5.1	100 ± 0.0	100 ± 0.0	100 ± 0.0	100 ± 0.0
80 °C	Heated, 2 h		5.1	100 ± 0.0	100 ± 16.7	100 ± 0.0	100 ± 0.0
100 °C	Heated, 30 min		5.1	83.3 ± 16.7	83.3 ± 16.7	100 ± 0.0	100 ± 0.0
pH							
2.0	Exposure to pH 2 at 37 °C, 2 h		6.8	50 ± 0.0	50 ± 0.0	50 ± 0.0	50 ± 0.0
4.0	Exposure to pH 4 at 37 °C, 2 h		6.8	50 ± 0.0	50 ± 0.0	50 ± 0.0	50 ± 0.0
6.0	Exposure to pH 6 at 37 °C, 2 h		6.8	50 ± 0.0	25 ± 0.0	50 ± 0.0	50 ± 0.0
8.0	Exposure to pH 8 at 37 °C, 2 h		6.8	50 ± 0.0	25 ± 0.0	50 ± 0.0	50 ± 0.0
Enzyme							
Proteinase K	2 mg/mL at 37 °C, 2 h		5.1	100 ± 0.0	66.7 ± 16.7	100 ± 0.0	100 ± 0.0
Trypsin	1 mg/mL at 37 °C, 2 h		5.1	83.3 ± 16.7	66.7 ± 16.7	83.3 ± 16.7	83.3 ± 16.7

Each value is presented as the mean ± standard deviation.

## Data Availability

The datasets generated and/or analyzed during the current study are available from the corresponding author on reasonable request.

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
