# Peer review of "Antimicrobial Activity of *Ligilactobacillus animalis* SWLA-1 and Its Cell-Free Supernatant against Multidrug-Resistant Bacteria and Its Potential Use as an Alternative to Antimicrobial Agents"

_microorganisms, 2023, doi:10.3390/microorganisms11010182_

Round 1

Reviewer 1 Report

The authors have performed a preliminary study to analyze the antimicrobial effect of L. animalis byproducts against MDR pathogens. There is few more information required in the manuscript;

1.       What was the source of L. animalis 11-2. The authors mentioned it has “which has relatively weak antimicrobial activity”.  What is the meaning of relative here? The authors should mention this in the results and not in the methods. A suitable reference is also required for this statement. 

2.       Please include the source of L. animalis SWLA-1 in the methods and how it was isolated.

3.       What is the difference between Ra and R? The authors need to define it in the manuscript.

4.       What is the cytotoxic effect? The authors need to provide the cytotoxic effect of the tested concentrations. It is important since a 10-fold concentration was used for CFS.

5.       What is the purpose of testing such a lower pH and a higher temperature? How it is related to the application?

Author Response

The authors have performed a preliminary study to analyze the antimicrobial effect of L. animalis byproducts against MDR pathogens. There is few more information required in the manuscript;

  1. What was the source of L. animalis 11-2. The authors mentioned it has “which has relatively weak antimicrobial activity”.  What is the meaning of relative here? The authors should mention this in the results and not in the methods. A suitable reference is also required for this statement. 
    -> We added the source of L. animalis 11-2 and revised the sentence following your comments in line 84-85.
  2. Please include the source of L. animalis SWLA-1 in the methods and how it was isolated.
    -> We added the source of L. animalis SWLA-1 and the reference containing isolation method in line 83.
  3. What is the difference between Rand R? The authors need to define it in the manuscript.
    -> We provided the meaning of Rin the footnote of Supplementary table, which is abbreviation of ‘Resistant’. For this reason, the first ‘R’ was written with superscription and the others were written just as ‘R’. We added the meaning of the abbreviation for R, S, and I in the manuscript in line 91-93.
  4. What is the cytotoxic effect? The authors need to provide the cytotoxic effect of the tested concentrations. It is important since a 10-fold concentration was used for CFS.
    -> We investigated the characteristics of the CFS depending on its concentration, not the cytotoxic effect. The cytotoxic effect of the CFS should be evaluated after we determine administration routes or challenge model of target animals in further study. In addition, the safety of L. animalis as a feed additive in animal was proved in previous study - Safety and efficacy of a feed additive consisting on Ligilactobacillus animalis ATCC PTA-6750 (formerly Lactobacillus animalis) for all animal species, EFSA Panel on Additives and Products or Substances used in Animal Feed, 2019.
  5. What is the purpose of testing such a lower pH and a higher temperature? How it is related to the application?
    ->We investigated the characteristics of CFS derived from L.animalis SWLA-1 are stable under various environmental change. We have evaluated the stability of the CFS under heat treatment considering it could be sterilized before used as a disinfectant or a feed additive. The stability under pH change or protease treatment was also evaluated to verify the CFS could be stable under the gastrointestinal environment when it is administrated per oral. These contents are added in the manuscript in line 375-378.

Reviewer 2 Report

I would like to thank the authors for their efforts to do this study which describing the antimicrobial activity of Ligilactobacillus animalis SWLA-1 against pathogenic multi-drug resistance bacteria isolated from human and veterinary samples. The experiment included a reference strain plus two positive antibiotics amikacin [AK] and ampicillin [AMP], in addition to test antimicrobial activity of the cell-free supernatant.

Major comments:

Study antimicrobial activity of new microbial isolates is very interested research point to avoid problems associated with multi-drug resistance. So, recent research in this field considers identifying natural products, anti-virulence factors, or mechanisms responsible for that action…etc. All the recorded data in the current study were reported as the size of clear zones or CFU count. I recommend additional data describing the structure of bacterial metabolites responsible for antimicrobial action and to purify and identify active compounds.

The current study mentioned the effect of different pH on indicator bacteria. However, many other environmental parameters such as temp, inoculum size, medial components…etc., should be analysed.

To avoid redundant or boring writing style, if the study of these environmental parameters will be done using one-factor at a time, the data should be moved to supplementary section; otherwise, using modern statistical analysis such as surface-response analysis will be more suitable to shorten the presented results.

The current study analyzed the effect of pH at time course on indicator bacteria to identify the optimum pH value to conduct the next experiments. However, I noted that the next experiments were conducted on different pH values. This point should be checked and discussed.

Presented results in table 2 are difficult to follow the significant differences between different treatments.

Table 3 also contains different treatments, and the style to show the significant variation between different treatments should be considered using scientific style avoiding redundancy.

L335-340 “Considering that the pH values of cocultured media between 0 h and 8 h ranged between 6.8 and 6.5, the significant inhibitory activity of L. animalis SWLA-1 resulted from other substances and not from acid stress. After 24 h of coculture, the pH dropped to less than 5.0. This made it difficult to evaluate the inhibitory activity of lactobacilli against the indicator bacteria without acid stress because, based on the results above, a pH of less than 5.5 is significantly harsh for bacterial strains.”  This is very interested challenge need to be addressed through the current study by conducted an experiment including 2 factors (pH and LAB) and the results should be analysed by 2-way ANOVA.

The current study mentioned many time the novelty of SII index based on longest and shortest diameters of inhibitory zones. The mathematical equation that was mentioned in the paper was used by other way in different study. I recommend checking that and discuss what is the novelty of calculated SII.

It was unclear how this study investigated the nature of antimicrobial substance of CFS. L352. “The nature of the antimicrobial substance in CFS derived from L. animalis SWLA-1 was investigated in this study.” However, the component of CFS was not determined.

What is the aim of current study to test the effect of different treatments such as high temperatures, pH neutralization, and enzyme on CFS activity?

Also the main idea behind the study of Comparative coculture assay need more clarification and discussion.

Author Response

  1. Study antimicrobial activity of new microbial isolates is very interested research point to avoid problems associated with multi-drug resistance. So, recent research in this field considers identifying natural products, anti-virulence factors, or mechanisms responsible for that action…etc. All the recorded data in the current study were reported as the size of clear zones or CFU count. I recommend additional data describing the structure of bacterial metabolites responsible for antimicrobial action and to purify and identify active compounds.
    -> In this study, we intend to prove significantly strong antimicrobial activity of L. animalis SWLA-1 and its CFS against MDR pathogenic bacteria with multiple experiments. The procedure for identification and purification of the active compounds in CFS is such large volume that is difficult to be merge in this study. We are planning to perform complete genome analysis of L. animalis SWLA-1 using NGS technique and other experiments for identification of the active compounds in CFS in further study. Purification method of the active compounds is also investigated and established in next study.
  2. The current study mentioned the effect of different pH on indicator bacteria. However, many other environmental parameters such as temp, inoculum size, medial components…etc., should be analysed.
    -> In this study, we used equal volume of same media (TSB-MRS broth), equal temperature and incubating condition, and equal inoculum volume of indicator bacteria using broth culture method. The variable was different pH only per each indicator bacteria in the experiment.
  3. To avoid redundant or boring writing style, if the study of these environmental parameters will be done using one-factor at a time, the data should be moved to supplementary section; otherwise, using modern statistical analysis such as surface-response analysis will be more suitable to shorten the presented results.
    -> Considering that the difference of pH between experimental groups is only variable factor, We have deleted the Table 1 in the manuscript and moved to supplementary section (Supplementary Table 2.) following your comments.
  4. The current study analyzed the effect of pH at time course on indicator bacteria to identify the optimum pH value to conduct the next experiments. However, I noted that the next experiments were conducted on different pH values. This point should be checked and discussed.
    -> We figured out that the growth of indicator bacteria was not affected by acid stress under pH range between 6.8-6.5 in this study. Based on this result, the next experiment—coculture assay, we mostly observed the antimicrobial activity of LABs against indicator bacteria in the same range.
  5. Presented results in table 2 are difficult to follow the significant differences between different treatments.
    -> We deleted the asterisk and presented numerical data only following your comment.
  6. Table 3 also contains different treatments, and the style to show the significant variation between different treatments should be considered using scientific style avoiding redundancy.
    -> We intended to show that the antimicrobial characteristics of the CFS derived from L. animalis SWLA-1 stay active under various treatment in this study. Following your comment, we are going to figure out he significant variation between different treatments after identification and purification of active compounds or molecules in further study.
  7. L335-340 “Considering that the pH values of cocultured media between 0 h and 8 h ranged between 6.8 and 6.5, the significant inhibitory activity of L. animalis SWLA-1 resulted from other substances and not from acid stress. After 24 h of coculture, the pH dropped to less than 5.0. This made it difficult to evaluate the inhibitory activity of lactobacilli against the indicator bacteria without acid stress because, based on the results above, a pH of less than 5.5 is significantly harsh for bacterial strains.”  This is very interested challenge need to be addressed through the current study by conducted an experiment including 2 factors (pH and LAB) and the results should be analysed by 2-way ANOVA.
    -> we had performed 2-way ANOVA and added the contents following your comment in line 211-214, and 338-341.
  8. The current study mentioned many time the novelty of SII index based on longest and shortest diameters of inhibitory zones. The mathematical equation that was mentioned in the paper was used by other way in different study. I recommend checking that and discuss what is the novelty of calculated SII.
    -> In this study, the calculated SII focuses on the ratio of inhibitory zone to the colony size not the differences between measured diameter of inhibitory zone and bacterial colony in other studies. Regarding the fact that 1mm of diameter could decide strong or less antimicrobial activity, the SII could help more precise comparison between multiple samples compared with absolute differences between diameters of inhibitory zone and colony. We added this explanation in manuscript in line 360-363
  9. It was unclear how this study investigated the nature of antimicrobial substance of CFS. L352. “The nature of the antimicrobial substance in CFS derived from L. animalis SWLA-1 was investigated in this study.” However, the component of CFS was not determined.
    -> we revised the sentence considering your comments in line 355-358.
  10. What is the aim of current study to test the effect of different treatments such as high temperatures, pH neutralization, and enzyme on CFS activity?
    -> ->We investigated the characteristics of CFS derived from L.animalis SWLA-1 are stable under various environmental change. We have evaluated the stability of the CFS under heat treatment considering it could be sterilized before used as a disinfectant or a feed additive. The stability under pH change or protease treatment was also evaluated to verify the CFS could be stable under the gastrointestinal environment when it is administrated per oral. These contents are added in the manuscript in line 375-378.
  11. Also the main idea behind the study of Comparative coculture assay need more clarification and discussion.
    -> we added several sentences to address main idea of comparative coculture assay in this study following your comment in line 326-329.

Round 2

Reviewer 1 Report

The authors have addressed all the questions.

Author Response

We already have responded all check points mentioned by reviewer 1.

Thank you for valuable reviewing and comments.

Reviewer 2 Report

After the second revision, there was not a big change between the two versions.

Major comments:

All the recorded data in the current study were reported as the size of clear zones or CFU count. I recommend additional data describing the structure of bacterial metabolites responsible for antimicrobial action and to purify and identify active compounds.

Using modern statistical analysis such as surface-response analysis will be more suitable to shorten the presented results.

Some results should be analysed by 2-way ANOVA. Although, the authors mentioned that you used 2-Way ANOVA, but there is no any feedback on the significance of interactions between factors.

The mathematical equation that was mentioned in the paper was used by other way in different study. I recommend checking that and discuss what is the novelty of calculated SII. What is the original reference of that equation.

Author Response

  1. All the recorded data in the current study were reported as the size of clear zones or CFU count. I recommend additional data describing the structure of bacterial metabolites responsible for antimicrobial action and to purify and identify active compounds.

-> We have dedicated to fulfilling the point you mentioned, but it is too tough to identify and purify the active compounds and figure out the structure of responsible metabolites within 10 days. These contents should be dealt with in further study.

  1. Using modern statistical analysis such as surface-response analysis will be more suitable to shorten the presented results.

-> We have performed surface-response analysis and added the result in line 217-221, 226-231 and Figure 2.

  1. Some results should be analysed by 2-way ANOVA. Although, the authors mentioned that you used 2-Way ANOVA, but there is no any feedback on the significance of interactions between factors.

-> We added the results of 2-Way ANOVA in line 288-297.

  1. The mathematical equation that was mentioned in the paper was used by other way in different study. I recommend checking that and discuss what is the novelty of calculated SII. What is the original reference of that equation.

-> In previous studies, the difference of diameter length between inhibition zone and the bacterial colony was major point to prove antimicrobial effect of bacteria. We focused on the ratio of diameter of bacterial colony to diameter of inhibition zone, not the difference of length. Furthermore, we validated this method by comparing the SIIs of lactobacilli with antibiotic disk that the bacteria are susceptible or resistant to. This procedure was performed to prove that quantitative comparison of SII between experimental groups is significant, not using the method already someone created. We mentioned this information in line 380-383.